# Adipose-Derived Stromal Cells within a Gelatin Matrix Acquire Enhanced Regenerative and Angiogenic Properties: A Pre-Clinical Study for Application to Chronic Wounds

**DOI:** 10.3390/biomedicines11030987

**Published:** 2023-03-22

**Authors:** Nicolo Costantino Brembilla, Ali Modarressi, Dominik André-Lévigne, Estelle Brioudes, Florian Lanza, Hubert Vuagnat, Stéphane Durual, Laurine Marger, Wolf-Henning Boehncke, Karl-Heinz Krause, Olivier Preynat-Seauve

**Affiliations:** 1Department of Pathology and Immunology, Faculty of Medicine, University of Geneva, 1205 Geneva, Switzerland; 2Division of Dermatology and Venereology, Geneva University Hospitals, 1205 Geneva, Switzerland; 3Division of Plastic, Reconstructive and Aesthetic Surgery, Geneva University Hospitals, 1205 Geneva, Switzerland; 4Laboratory of Therapy and Stem Cells, Geneva University Hospitals, 1205 Geneva, Switzerland; 5Program for Wounds and Wound Healing, Geneva University Hospitals, 1205 Geneva, Switzerland; 6Laboratory of Biomaterials, Faculty of Dental Medicine, University of Geneva, 1205 Geneva, Switzerland; 7Department of Medicine, Faculty of Medicine, University of Geneva, 1205 Geneva, Switzerland

**Keywords:** adipose stem cells, clinical translation, mesenchymal stem cells, stromal cells

## Abstract

This study evaluates the influence of a gelatin sponge on adipose-derived stromal cells (ASC). Transcriptomic data revealed that, compared to ASC in a monolayer, a cross-linked porcine gelatin sponge strongly influences the transcriptome of ASC. Wound healing genes were massively regulated, notably with the inflammatory and angiogenic factors. Proteomics on conditioned media showed that gelatin also acted as a concentrator and reservoir of the regenerative ASC secretome. This secretome promoted fibroblast survival and epithelialization, and significantly increased the migration and tubular assembly of endothelial cells within fibronectin. ASC in gelatin on a chick chorioallantoic membrane were more connected to vessels than an empty sponge, confirming an increased angiogenesis in vivo. No tumor formation was observed in immunodeficient nude mice to which an ASC gelatin sponge was transplanted subcutaneously. Finally, ASC in a gelatin sponge prepared from outbred rats accelerated closure and re-vascularization of ischemic wounds in the footpads of rats. In conclusion, we provide here preclinical evidence that a cross-linked porcine gelatin sponge is an optimal carrier to concentrate and increase the regenerative activity of ASC, notably angiogenic. This formulation of ASC represents an optimal, convenient and clinically compliant option for the delivery of ASC on ischemic wounds.

## 1. Introduction

Chronic skin wounds affect about 1–2% of the worldwide population, and up to 5% of subjects older than 65 years [1,2]. The treatment landscape for the management of chronic wounds spans from dressings, debridement, negative pressure to advanced skin replacement technologies [3,4]. While the efficacy of each of these therapies has been shown in specific settings, a unique gold-standard in chronic wound management is lacking. Even in the presence of guidelines, severe ulcers are not efficiently managed with current therapies. In this respect, cell therapy based on adipose-derived stromal cells (ASC) has emerged as particularly promising [5,6]. Several in vitro and pre-clinical animal studies have demonstrated that ASC exert beneficial effects on wounds [5,7,8,9,10]. ASC promote healing [11]; suppress excessive inflammatory responses [12,13,14]; increase survival and proliferation of fibroblasts allowing ExtraCellular Matrix (ECM) remodeling [15,16]; produce anti-fibrotic factors [17]; and promote neovascularization [18]. The therapeutic effects of ASC were shown to depend mainly on paracrine mechanisms and production of extracellular vesicles [19,20]. Several human clinical trials have also been reported worldwide, most of them of limited reliability (i.e., uncontrolled studies). The appearance in recent years of a few well-conducted controlled studies has provided a new perspective to interpret the efficacy of ASC-based therapies for patients with chronic wounds [5]. Some recent and controlled trials reported that expanded ASC or the stromal vascular fraction have a superior efficacy compared with the control group, with a satisfactory safety profile. The delivery method and ASC stability in vivo remain, however, major challenges that may compromise the large-scale application of ASC therapy. The most common modality used in past and on-going trials relies on multiple intramuscular, intra-wound or peri-wound injections [21,22]. This route of administration is not well controlled spatially and it is still unclear where the ASC should be injected among the possibilities between the dermis, the adipose tissue or the adjacent muscles [23]. This route is also very painful due to frequent local inflammation. Furthermore, in the absence of scaffolds, ASC do not have the opportunity to concentrate their regenerative secretome and be regulated by their environment in a way that promotes wound healing. In addition, suspension-delivered ASC have been shown to be locally unstable due to cell migration or death, or to be rapidly trapped in the lungs before reaching the wound [24,25,26]. Collectively, these limitations due to a lack of knowledge and control of ASC delivery routes still limit standardization and progress toward a clinical reality [8,10].

In this study, we evaluated the introduction of ASC within a clinical-grade surgical sponge composed of crosslinked porcine gelatin. This formulation not only aimed at concentrating locally the ASC regenerative secretome in the wound bed, but took advantage of the pro-healing and adsorption properties of gelatin. The impact of crosslinked gelatin on ASC within this specific environment was notably investigated and showed strong regulations of the healing properties, notably angiogenesis. The ASC delivery studied here represents an easy and convenient formulation to limit the constraints of the current injection-based protocol of ASC therapy for chronic wounds.

This study demonstrates that the introduction of ASC into cross-linked porcine gelatin sponges strongly influences their biological activity, in particular by regulating numerous genes involved in the wound healing process. In particular, genes related to angiogenesis were increased. When applied topically to ischemic rat wounds, the enhanced angiogenic properties of ASC by the gelatin sponge were confirmed, facilitating revascularization and wound closure more rapidly than standard treatments. Thus, cross-linked gelatin scaffolds represent a convenient, biocompatible, and effective delivery route for concentrating ASC and increasing their wound healing capacity.

## 2. Materials and Methods

### 2.1. ASC Culture and Engineering of an ASC-Enriched Patch

Human ASCs were prepared from the non-ischemic subcutaneous fat of donors. The ASC lines used in this study were fully validated for their phenotype, multipotency and regenerative potential. ASC were used between passage 2–5 and were cultured in Dulbecco’s Modified Eagle Medium DMEM (4.5 g/L glucose, L-Glutamine) supplemented with 10% of human platelet lysate (Stemulate, Cook Regentek, Bloomington, IN, USA), 1% penicillin and streptomycin (ThermoFisher, Waltham, MA, USA) at 37 °C and under 5% CO_2_. Rat ASC were grown in the same medium supplemented with 10% of fetal calf serum (ThermoFisher, Waltham, MA, USA). To manufacture the ASC-gelatin sponge, a piece of sterile absorbable gelatin sponge USP Gelfoam (Pfizer, Brooklyn, NY, USA) was soaked in a suspension of ASC at a final density of 6000 cells/mm^3^.

### 2.2. Flow Cytometry and Multipotent Differentiation of ASC

Cells were incubated with fluorochrome-labeled antibodies for 30 min at 4 °C in binding buffer (BD Biosciences, Allschwil, Switzerland), prior to analysis using a BD Accuri^TM^-B6 flow cytometer (BD Biosciences, Allschwil, Switzerland). Antibodies were as follows: (i) for human cells: mouse IgG1 anti-CD44/CD73/CD90/CD45/CD105/CD14/HLA-DR (all from Abcam, Cambridge, UK); (ii) for rat cells: Armenian hamster anti-rat CD29-APC (clone HMb1-1, ThermoFisher, Waltham, MA, USA), mouse anti-rat-CD31-PE (clone TLD-3A12, BD Biosciences, Allschwil, Switzerland), mouse anti-rat CD45-BV421 (clone OX-1, BD Biosciences, Allschwil, Switzerland) and mouse anti-rat CD90-BB515 (clone OX-7, BD Biosciences, Allschwil, Switzerland). Analysis was performed on viable cells (negative for Draq7) upon exclusion of cell doublets. Cell purity was >98%. The multipotent differentiation into adipocytes, osteocytes and chondrocytes was performed by using the Human Mesenchymal Stem Cell Functional Identification Kit (R&D Systems, Minneapolis, MN, USA) according to the supplier’s instructions.

### 2.3. Immunocytochemistry and Immunofluorescence on Tissue Sections

ASC were cultured on glass coverslips prior to fixation with paraformaldehyde 0.5% for 15 min at RT. Cells were incubated overnight (o/n) at 4 °C in PBS containing 0.3% Triton X-100 and 0.5% bovine serum-albumin with the following primary antibodies: mouse IgM anti-Stro-1 (Clone STRO1, ThermoFisher, Waltham, MA, USA). Detection was achieved using an anti-mouse IgM-Alexa 555 antibody for one hour at +4 °C. Cells were stained with DAPI 1 μg/mL for 10 min prior to final washing and mounting. For histological analyses, tissues were washed in PBS and fixed with a 4% paraformaldehyde solution for 20 min prior to dehydration and embedment in paraffin. Upon rehydration, slides were stained in PBS supplemented with bovine serum albumin 1%, Triton X-100 0.3% o/n at 4 °C with a mouse IgG anti-CD31 (Abcam, Cambridge, UK). Upon staining with anti-mouse IgG-Alexa 555 antibody, slides were counterstained with DAPI and mounted in FluorSave medium (Calbiochem, Buchs, Switzerland). Hematoxylin and Eosin staining and Masson’s trichrome staining were performed according to the standard protocol. Vessel area was computed by QuPath software as a function of CD31 staining (above a defined threshold) in at least 3 sections per condition analyzed. Immunocytochemistry and immunofluorescence applied to osteocytes, chondrocytes and adipocytes derived from ASC was performed with the reagents of the human mesenchymal stem cell functional identification kit (RnDSystems, Minneapolis, MN, USA).

### 2.4. Cytokine Measurements

Cytokines were measured in the supernatants from ASC cultures and ASC-enriched patches using the human cytokine base kit A (RnDSystems, Minneapolis, MN, USA) combined with a magnetic Luminex assay (Bio-plex 200, Biorad, Hercules, CA, USA) according to the manufacturer’s instructions.

### 2.5. Transcriptomic

A microarray was used as the best way to simply analyze the global cell regulation within a gelatin sponge environment. Isolation of total RNA was performed by using RNeasy kit from Qiagen (Hombrechtikon, Switzerland) according to the manufacturer’s instructions. RNA concentration was determined by a spectrometer (Thermo Scientific™ NanoDrop 2000, ThermoFisher, Waltham, MA, USA) and RNA quality was verified by 2100 bioanalyzer (Agilent, Santa Clara, CA, USA). Human and rat microarray was performed with the Clariom^TM^ S Assay’s for human and rat (ThermoFisher, Waltham, MA, USA), respectively, using the Complete GeneChip^®^ Instrument System, Affymetrix. Hierarchical clustering and principal component analysis were computed using TAC4.0.1.36 software (Biosystems, Muttenz, Switzerland) with default settings. Gene Set Enrichment Analysis (GSEA) was used to analyze the pattern of differential gene expression between the human ASC-patch and the monolayer condition. The Gene Ontology Biological Process (GOBP) gene set from the Molecular Signatures Database was used. The results of GSEA analysis were visualized for enrichment map using Cytoscape 3.8.2, (Moutain view, CA, USA). The enrichment of processes and pathways within the significantly upregulated or downregulated transcripts (fold change > 2, FDR < 0.01) identified in the rat ASC-path compared to rat ASC grown in monolayer was assessed using Metascape (www.metascape.org, accessed on 11 March 2022). The parameters used for the analysis were as follows: Organisms: Rattus Norvegicus, Input gene set: GO Biological Process; Min Overlap: 3; *p* value cutoff: 0.01; Min enrichment: 0.01.

### 2.6. Mass Spectrometry

Cultured human ASC or ASC-enriched patches were incubated for 45 min with collagenase NB6 (Nordmark, Uetersen, Germany) at 0.3 U/mL, washed with a serum-free DMEM (ThermoFisher, Waltham, MA, USA) and cultured for 24 h at 37 °C in serum-free medium. Upon clarification of supernatants at 500× *g* for 10 min, proteins were precipitated, digested and peptides analyzed by nanoLC-MSMS using an easynLC1000 (ThermoFisher, Waltham, MA, USA) coupled with a Q Exactive HF mass spectrometer (ThermoFisher, Waltham, MA, USA). Database searches were performed with Mascot (Matrix Science, London, UK) using the Human Reference Proteome database (Uniprot). Data were analyzed and validated with Scaffold (Proteome Software, Portland, OR, USA) with 1% of protein FDR and at least 2 unique peptides per protein with a 0.1% of peptide FDR.

### 2.7. Chick Chorioallantoic Membrane Model

To estimate the in vivo angiogenic properties of a gelatin/ASC patch, the CAM model was a first simple and rapid experimental approach. Fertilized chicken eggs were incubated at 37 °C and placed with the smaller convexity pointing upward from ED1 (Embryo Development day) to ED4. At ED4, a hole was drilled through the smaller convexity pointing of the shell. At ED 7, the eggs were opened with scissors through the hole and the inner membrane to create a round window with approximate 1 cm diameter. The developing chorioallantoic membrane was then irritated through creation of a micro-hemorrhage. With ASC in suspension, a silicon ring with a 4 mm inner diameter was placed on the site of the generated hemorrhage. The ASC-enriched patched, fibroblast-enriched patches or control empty patches were deposited directly on the site of the generated hemorrhage. After implantation, the window in the eggshell was covered with a paraffin film and placed in the incubator at 37 °C. The number of vessel connections to the patch were counted manually under a binocular loop.

### 2.8. Migration and Tubulogenesis of HUVEC

Migration and tubulogenesis of HUVEC was an efficient way to discover the functional influence of the ASC/gelatin secretome on endothelial cells. HUVEC were purchased and cultured in complete endothelial cell medium 2 (both from Sigma, Buchs, Switzerland). Migration of HUVEC was analyzed by using the endothelial cells migration assay (Sigma, Buchs, Switzerland) according to the manufacturer’s instructions. Briefly, HUVEC were starved for 15 h in the endothelial cell medium 2 without serum and supplement and introduced in a Boyden chamber with a hemi-permeable membrane coated with fibronectin or Bovine Serum Albumin (BSA) used as a control at the bottom. Migration through the fibronectin layer towards supplement-free endothelial cell medium 2 conditioned 48 h by ASC was measured by cell coloration (crystal violet) and extraction of the dye having migrated outside the Boyden chamber (via measurement of the absorbance of the extract at 540 nm). The migration was calculated as the difference between the absorbance with fibronectin and the absorbance with control BSA. For the tubulogenesis assay, serum/supplement-free endothelial cell medium 2 was conditioned for 48 h with ASC. The analysis of tubular assembly of HUVEC was performed in the presence of each conditioned medium by using the angiogenesis assay kit (Abcam, Cambridge, UK) according to the manufacturer’s instructions. Briefly, HUVEC were plated in their conditioned medium on a fibronectin-containing gel for 24 h, prior to cell coloration by a fluorescent dye and analysis of tubes via the Cytation 5 cell imaging reader (Agilent, Santa Clara, CA, USA).

### 2.9. Animal Experiments

For stability/tumorigenicity assays, the method recommended by the European Pharmacopoeia (EMEA/149995/2008) was used. ASC-enriched patches or Hela cells (5 × 10^5^ cells) were subcutaneously transplanted in the right flank of 10 Nu/Nu mice, followed for 12 weeks for tumor palpation and necropsy. At week five, 9 out of 10 mice that received Hela cells developed palpable tumors, confirming the validity of the test. The model of ischemic wound in the rat was the best available animal model of ischemic wounds and performed as previously described [25,27]. Wistar female rats of 250–300 g were pre-anesthetized by inhalation of isoflurane 5%, and anesthetized at the dose of 2%. Hairs were removed from the inguinal region using a mechanical shaver. All surgical procedures were performed under an operating microscope. Through a longitudinal incision made in the upper part of the left thigh, the external iliac and femoral arteries were dissected free along their entire length, from the common iliac to the saphenous artery, and one cm-length artery was removed. Immediately after the arterial resection, a wound was created on the dorsal aspect of the feet in all animals by removing a full-thickness skin area of 1.2 × 0.8 cm. Treatments were applied a day after the surgery. Rat ASC were generated from the inguinal non-ischemic fat of control rats two months after the induction of paw ischemia. Treatments were applied a day after the wound creation. To maintain the patches on the wound, a gutter of perforated silicone interface (Mepitel, Molnlycke, Singapore, Singapore) was covered with a thin sheet of polyurethane (Opsite, SmithNephew, London, UK) and sutured around the wound. The patches were removed at day 7 and the wound covered with polyurethane until full recovery. Daily macroscopic evaluation of the limbs and feet as well as wounds planimetry were performed until complete wound healing.

### 2.10. Statistical Analysis

Statistical analysis was performed using GraphPad Prism version 6.0 (Graphpad Software, La Jolla, CA, USA). *p*-values less than 0.05 were considered statistically significant, and were indicated as follows: *: *p* < 0.05; **: *p* < 0.01; ***: *p* < 0.001 (non-parametric Mann–Whitney *t* test).

## 3. Results

### 3.1. Culture of ASC in a Sponge Made of Porcine Crosslinked Gelatin

ASC were introduced and cultured in a sponge made of crosslinked porcine gelatin (Appendix A). It was indeed desirable to study a scaffold whose structure, functions and mechanical properties are similar to those of healthy skin and compatible with healing [28]. A number of scaffolds have been widely used in the field of tissue engineering and all emphasize the importance of hydrophilicity, biodegradability and biocompatibility. Collagen is an important component of the skin and provides strength. Gelatin is a hydrophilic, biocompatible, and inexpensive collagen-derived product and was considered to have the desired characteristics to promote ASC survival, adhesion, and activity. Several ASC lines used in this study were prepared from the adipose tissue of donors. The ASC identity was confirmed by their phenotype (CD44+/CD73+/CD90+/CD45−/CD105+/CD14−/HLA-DR−) and ability to be differentiated towards chondrocytes, osteoblasts and adipocytes under appropriated differentiation conditions and according to international standards [29]. Inter-donor variability, as assessed by computing the coefficient of variation of the whole transcriptome among the different ASC lines, was minimal (mean ± SD of 5.7 ± 4.0%). The generated ASC-gelatin sponge was easy to handle (Figure 1A, left), and characterized histologically by a dense cellular tissue (purple) interspersed between gelatin trabeculae (dark pink) (Figure 1A, middle). A green halo around gelatin trabeculae suggested collagen dissolution as assessed by Masson’s trichrome staining (Figure 1A, right). ASC that clustered in pores organized into a compact tissue composed of collagen fibers from their own secretion. This tissue resulted from an ASC-dependent secretion of ECM and contraction of the gelatin sponge, as only isolated cells in a rarefied gelatin mesh were observed at the beginning of the culture (Appendix A). A low cellular density was instead obtained upon parallel culture of ASCs within a conventional collagen gel (Appendix A). ASC retained their stromal identity within the gelatin sponge, as shown by sustained and widespread expression of the stromal marker Stro-1 (Figure 1B). Furthermore, cells obtained upon enzymatic dissociation of the ASC-enriched patch had a transcript profile compatible with undifferentiated ASC as defined by current guidelines [30] (Figure 1C). Confirming the ASC identity, cells extracted from the ASC-enriched patch retained multipotent capacity, being able to differentiate towards osteoblasts (expressing osteocalcin), adipocytes (FAB4) and chondrocytes (aggrecan) in appropriate culture conditions (Figure 1D). Thus, ASC could efficiently be included and grown within a clinical-grade and crosslinked porcine gelatin sponge to generate an undifferentiated ASC-enriched compact tissue, which has physical properties compatible with its use as a cellular patch.

### 3.2. ASC in a Gelatin Sponge Enhanced Their Regenerative Transcriptome

We next investigated whether ASC have modified their gene expression capabilities because of their interaction with porcine gelatin in spongious conditions. To this aim, the transcriptome of cells extracted from three independent ASC-gelatin sponges was compared with ASC from the same batch, but grown in parallel in monolayers. This latter condition was performed in line with the standard protocol used to produce ASC for injection-based ASC therapy [8]. Hierarchical clustering and principal-component analysis revealed that ASC strongly modified their global transcriptome when cultured within the gelatin sponge, compared to ASC in monolayers (Appendix A, respectively). ASC grown in monolayers for 7 days maintained a transcriptome profile-like cells prior to culture (freshly isolated ASC vs. monolayer (d7), Appendix A). ASC within the patch did not differentiate in fibroblasts nor embryoid bodies (Appendix A). To explore the characteristics of the genes expressed in ASC-patches compared to ASC grown in monolayers, we performed a threshold-free gene set enrichment analysis (GSEA). The 50 most highly differentially expressed genes (top 25 upregulations and top 25 downregulations) are shown in Figure 2A. The most up-regulated genes and pathways were linked to critical components of the healing process, namely immune function, morphogenesis, and vascular growth. Down-regulations were linked to DNA regulation and mitochondrial functions (Figure 2B). Smaller clusters are shown in Appendix A. Analysis of the 191 genes reported to be most implicated in the wound regeneration process (list available upon request) resulted in the identification of 51 significantly regulated transcripts (fold change > 2, *p* < 0.005): 30 strongly up-regulated (4× to 60×), 13 minimally up-modulated (<4×), five minimally down-modulated (<4×) and three modestly down-regulated (4× to 10×) (Figure 2C). The strongest up-regulated transcripts observed were CXCL8 (IL-8) (potent angiogenic, chemotactic and inflammatory cytokine), CXCL6 (angiogenic, chemotactic, anti-microbial), IL-6 (angiogenic, pro-inflammatory), and CXCL5 (angiogenic, matrix remodeling factor, pro-inflammatory). Other angiogenic factors were up-regulated (VEGF, ANGPTL2, ADM), as well as matrix remodeling factors (MMP16), collagens, cell growth factors (HGF) and several chemokines. Keratinocytes Growth factor (KGF or FGF-7), promoting epithelialization during skin wound healing, was similarly upregulated. Together, these observations show that the interaction of ASC with gelatin within resulted in a general enhancement of their regenerative transcriptome.

### 3.3. The Secreted Proteome of ASC Is Absorbed by Gelatin

The composition of the proteome secreted from ASC in a gelatin sponge was profiled by mass spectrometry and compared to that of control ASC grown in monolayers, or the empty sponge. Hierarchical clustering analysis allowed for the distinction of the three conditions (Figure 3A). The number of proteins found in the supernatants from ASC-gelatin was less than the sum of the proteins found in the other two conditions (Figure 3B), suggesting protein adsorption within the patch. Accordingly, the protein diversity was slightly higher in supernatants from ASC grown in monolayers if compared to ASC-gelatin (Figure 3A,C). Overall, 3.1% of proteins were down-regulated and 2.7% up-regulated in ASC-patches as compared to the monolayers’ supernatants (The complete list is available upon request). Quantification by bio-arrays of cytokines, growth and angiogenic factors having a key role in wound healing confirmed their relative concentration in the supernatant from ASC-patches (Appendix A). Analysis of the supernatants from empty patches revealed that the gelatin dissolved, releasing 49 proteins on average, most being collagens and keratins (Appendix A). We next assessed the ability of the gelatin of the patch to adsorb proteins secreted by ASC. Three empty gelatin sponges were incubated with a serum-free medium or a serum-free medium previously conditioned by ASC. Mass spectrometry analysis was performed on the mixture after complete dissolution of the sponge. Twelve vs. one hundred and eighty-one proteins were found in sponges exposed to an empty serum-free medium vs. an ASC-conditioned media, confirming that gelatin had adsorbed several factors produced by ASC (Figure 3D,E). Interestingly, fibronectin-1 (FN1), a molecule important for the healing process [31], was the most abundant protein derived from ASC adsorbed on gelatin. The complete list is available upon request. Together, these proteomic data indicate that the ASC-gelatin sponge actively released factors that are derived from both ASC and gelatin. In addition, our results indicate that the patch functioned as a reservoir of healing factors.

### 3.4. The ASC Gelatin Sponge Promotes Angiogenesis

Neo-vascularization being a most critical process for wound healing, we next investigated the ability of the ASC-gelatin sponge to promote angiogenesis in vivo by using the chicken chorioallantoic membrane (CAM) model (Figure 4A). ASC in gelatin were superior to all conditions tested, as a significantly higher number of vessel branches sprouting from the dressing were observed (Figure 4A). Notably, new blood vessels could only be visualized in the ASC-gelatin sponge by macroscopic analysis, but not the other conditions. A single cell suspension of ASC derived from standard culture (contained by a silicon ring) did not induce significant new vessel branches. The pro-angiogenic activity of the ASC-gelatin sponge was probably due to synergistic mechanisms between ASC and gelatin, since it was not merely the addition of the effect of its isolated constituents. In complementary in vitro experiments, conditioned media from the ASC-gelatin sponge increased the migration of Human Umbilical Endothelial Vein Cells (HUVEC) through a fibronectin-containing hemi-permeable membrane and better promoted tubular assembly of HUVEC as compared to media from the same number of ASC alone or empty sponge (Figure 4B). None of the condition tested interfered with HUVEC proliferation. Finally, healthy human keratinocytes seeded on top of the ASC-patch developed a fully stratified epidermis with an intact basement membrane (Appendix A), and healthy human fibroblasts exposed to ASC-patch conditioned media showed an increased in vitro survival compared to empty sponge-conditioned media (Appendix A). Together, we show that the ASC in a gelatin sponge harbor more angiogenic properties than ASC alone, promoting tubular assembly and endothelial cell migration, and does not inhibit the growth of key cutaneous cellular components.

### 3.5. ASC in Gelatin Sponge Stability and Tumorigenicity In Vivo

Since ASC in a gelatin sponge acquired a pro-angiogenic profile, we tested if they could promote tumorigenesis in immunodeficient nude mice. None of the 10 transplanted mice developed signs of palpable tumors near the transplant or in peripheral tissues, whereas 10 mice transplanted with Hela control cells developed tumors. Notably, all transplanted ASC-gelatin sponges showed macroscopic vascularization at week 5 (Figure 5A, left). Histological assessment of the transplanted ASC-patches revealed the presence of a tissue-like cell-rich gelatin mesh (Figure 5B). Culture of cells extracted upon enzymatic digestion of the transplanted ASC sponges had the morphology of ASC and expressed the Human Nuclear Antigen (HNA) (Figure 5C). Flow cytometric analysis of the secondary ASC line confirmed the maintenance of the ASC phenotype (positivity for CD73, CD90, CD105, and CD44 and absence of CD45, HLA-DR and CD14) (Figure 5D). Finally, the size of the transplanted ASC-patches was stable in vivo until week 3, and was completely resorbed by week 12 (Figure 5A, right).

Together, these in vivo experiments indicated that ASC in gelatin lacked any tumorigenic activity and allowed the stabilization of the ASC for at least 5 weeks, while the whole product was resorbed within 12 weeks in immunodeficient setting. The ASC-patch thus had essential pre-clinical safety requirements for further development in humans.

### 3.6. ASC in Gelatin Sponge Accelerates the Healing and Induces Early Neo-Angiogenesis in a Rat Model of Ischemic Wound

Next, the in vivo efficacy of ASC in a gelatin sponge in a pre-clinical rat model of an ischemic wound was investigated [25]. Wounds were created on the dorsal part of the hind paws of Wistar rats by removing a full-thickness skin area. To create paw ischemia, 1 cm of the femoral artery was surgically removed prior to the wound creation. In these experiments, ASC-patches were prepared from non-consanguineous rat ASC (rASC-patch). The identity of rat ASC was confirmed by flow cytometry. Rat ASC grown within the patch generated a dense tissue-like structure (Appendix A). Transcriptomics revealed the presence of differences in the rat ASC-patch as compared to rat ASC monolayer cultures (Appendix A). Metascape-based enrichment analysis using the GO-Biological Process gene set showed that the most enriched pathways and processes identified within the significantly up-regulated genes (fold change > 2, FDR < 0.05) included extracellular matrix deposition, response to growth factors and angiogenesis. Downregulations were instead linked to DNA regulation and cell cycle (Appendix A). These results are in line with the observations in humans, considering species-specific differences, and indicate that rat ASC enhance their regenerative transcriptome when grown within the patch. Of note, rats treated with rat ASC-patches healed faster than rats treated with empty gelatin patches or standard silicone/polyurethane dressings (Figure 6A,B). A granulation tissue was macroscopically visible from day 9 in the ASC-patch treated group, while tendons were still exposed in the control groups (Figure 6A). All rats treated with the ASC-enriched patches reached a complete wound closure before day 17, compared to 67% of rats treated with the empty gelatin sponge (Figure 6B). The complete wound closure was confirmed histologically (Figure 6C). In line with the observed effects on angiogenesis, staining for the endothelial marker CD31 revealed an increased vascularization in the healing tissue of rats treated with ASC-patches (Figure 6D). In this condition, angiogenesis was faster and sustained over time, leading to vessels characterized by a higher diameter and organized a denser network. Quantification of the total vessel area confirmed the superiority of the ASC-patch over the empty gelatin sponge and standard dressing (Figure 6D). Finally, we analyzed the ASC survival within the ASC-patch in vivo. Rat-ASC stably transduced with firefly luciferase (FLuc) under the control of the ubiquitous promoter EFS (rASC_EFS FLuc_) were used to generate rat ASC-patches (FLuc-rASC-Patch). Intraperitoneal injection of D-luciferin allowed the monitoring of ASC survival in vivo by using the live imaging system IVIS Spectrum (Perkin Elmer, Waltham, MA, USA). Luminescence in the FLuc-ASC-Patch was detected until removal of the treatment at day 7, confirming the survival of ASC in vivo (Figure 6E,F). Luminescence was maintained after treatment removal until day 17, indicating that some ASC were engrafted within the healing wound. Additional quantifications during the entire period of treatment indicated a fast increase of the luminescence until day 6, an observation in favor of rapid exchanges of the patch with the biological fluids and the increased neovascularization previously observed (Figure 6F). No ectopic tissue formation was observed outside the wound area (Appendix A). Together, we provide evidence that the ASC within the gelatin sponge were stable and viable in vivo for at least 17 days. Notably, the ASC-patch accelerates wound healing in a rat model of ischemic wounds, resulting in increased neo-vascularization and emergence of a granulation tissue earlier in the healing process.

## 4. Discussion

In this study, we demonstrate that ASC introduced within a porcine crosslinked-gelatin sponge not only concentrated their secretome, but also regulated their regenerative activity in favor of regeneration and angiogenesis. The pre-clinical efficacy/safety of a cellular patch composed of ASC in gelatin sponge is demonstrated in a preclinical model of ischemic wound healing. The adipose tissue was preferred to other sources for the preparation of mesenchymal stromal cells (MSC) due to its easier accessibility, which guarantees a higher translational relevance in possible clinical applications. Extensive molecular and biochemical characterization revealed that induction of angiogenesis, in the absence of tumorigenesis, is one of the most important mechanisms of action of this approach. The patch formulation represents an optimal non-invasive delivery route that maximizes the local effects of ASC within the wound. Compared to intra- or peri-wound injection of ASC, the ASC-gelatin sponge guarantees an extended local stability and viability of ASC and the absence of extra-wound migration. This study validates the patch approach in the pre-clinical setting and paves the way for its use in first-in-human studies. The effect of the ASC-patch mainly relies on the combined interaction of ASC with gelatin. ASC are known for their ability to enhance the healing process by secretion of soluble healing factors [32,33,34,35]. Here, we demonstrate that ASC within the patch not only increase their own regenerative potential, but also locally concentrate pro-healing factors. ASC exposed to gelatin microcryogels have been shown to increase their expression of VEGF, HGF, FGF, and PDGF [36]. The formation of a three-dimensional tissue organization within the patch enables the optimization of cell–cell interactions, paracrine events and gas exchange/oxygen supply [37,38]. The gelatin sponge itself may promote the healing process independently of ASC. Gelatin acts as a chemotactic agent [39], promotes the formation of a granulation tissue [40] and absorbs exudates present in the wound bed [41]. In line with this data, gelatin was shown to directly enhance wound closure [39], particularly by increasing angiogenesis, keratinocyte and fibroblast proliferation/migration and myofibroblast differentiation [42,43,44]. Interestingly, the local administration mesenchymal stem cells in a collagen scaffold led to better regeneration of soft gingival tissues in rabbits through enhanced gingival vascularization and epithelization with a clear positive correlation between vascular growth and epithelial response [45]. Additionally, epithelialization and angiogenesis were linked in another study where diabetic wounds received mesenchymal stem cells activated with LPS: the granulation tissue of treated wounds had higher pronounced epithelialization and associated vascularization compared with controls [46]. Besides the direct action of gelatin, we showed that the gelatin patch dissolves in aqueous conditions, releasing soluble collagens and cytokeratins known to favor wound healing through enhancement of proliferation/migration of fibroblasts and keratinocytes [47,48,49,50,51,52,53]. Previous studies have shown the possibility to include MSCs in acellular scaffolds in vitro [54]. A dermic substitute of collagen or atelocollagen loaded with MSC, was also tested in animals [55] or patients [56]. The efficacy was confirmed in an animal model considered to be relevant for clinical translatability, although not fully recapitulating all feature of the human disease. Of note, the in vivo experiments described in this study were performed in an allogeneic setting, as outbred non-consanguineous rats were used. Despite a head-to-head experiment is lacking, the ASC-patch approach promoted a faster wound closure when compared to ASC locally injected in peri-wound area, as assessed in our previous independent experiments performed using the same rat model [25]. The advantages of using a crosslinked gelatin as a scaffold for ASC delivery are the following: it is a regulatory approved, clinical-grade support to ensure an easier and faster clinical translation. Gelatin is indeed a biodegradable and biocompatible scaffold that is already widely used in clinical settings without any antigenicity/toxicity. Compared with collagens or atelocollagens, gelatin is cheaper than collagen, an essential feature for pharmaceutical development. It is also more hydrophilic, an important property ensuring the maintenance of a moist environment on the wound. The porosity of the sponge is particularly attractive to allow ASC integration, survival, simultaneously permitting the biological fluids to circulate, as well as the colonization by host cells and vessels. In line with this, we have observed a rapid irrigation of the sponge when applied on ischemic wounds and a colonization of the patch by host cells. The malleability and physical stability of gelatin sponges allows their spatial adaptation to the wound bed, an important prerequisite for clinical use. Finally, the adsorption and local concentration of the ASC secretome are beneficial to accelerate healing compared to cell suspensions. Thus, the gelatin-based delivery studied here represents an attractive, non-invasive route, which maximizes the local efficacy of ASC therapy for chronic wound care without toxicity/tumorigenicity. The risk of side effects of ASC/gelatin therapy are considered to be limited because (i) clinical-grade gelatin is widely used in surgery worldwide (as hemostatic sponges), and (ii) ASC have been extensively studied and administrated through many delivery routes, showing an excellent tolerance in humans [5]. First-in-human studies are planned soon to clinically validate the use of the ASC-patch solution as a simple and effective treatment improving patient compliance and physician acceptance. The possibility to use the ASC-patch in an allogenic setting in clinical practice is an option that still requires further validation. Indeed, an allogeneic approach would have several advantages over an autologous approach, including the use of batches of certified and validated ASC, reduced logistical constraints, greater scalability, and substantial cost reductions.

## 5. Conclusions

The scientific novelty of this study is the demonstration that a gelatin-based matrix modifies ASC in favor of the secretion of factors increasing wound healing and vascularization. This regulation is complemented by the ability of the matrix to locally concentrate and adsorb cell secretions for optimal delivery to the wound bed. This comprehensive preclinical study demonstrates the safety and efficacy of a gelatin-based stem cell patch and is a prerequisite for a future clinical study in humans.

## 6. Patents

NB, OPS, KHK and WHB are the inventors of patent application PCT/EP2020/076083 covering the culture of ASC within the gelatin support.

## Figures and Tables

**Figure 1 biomedicines-11-00987-f001:**
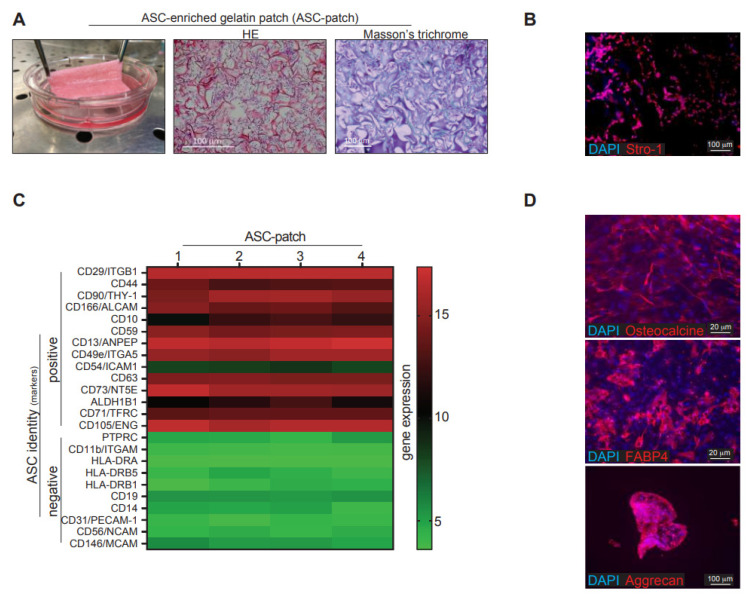
ASC cultured within a gelatin sponge generates a stable and easy-handling cellular patch (ASC-patch). (**A**) Macroscopic image of an ASC-patch (left). Hemalum/eosin coloration (middle) and Masson-trichrome staining (right) of an ASC-patch are shown. (**B**) Immunofluorescence staining of an ASC-patch section for the stromal marker Stro-1. (**C**) Full gene expression array of ASC-patches generated from 4 independent ischemic patients. Commonly used positive markers for ASC identification are shown. (**D**) Differentiated towards osteoblasts, adipocytes or chondrocytes of ASCs extracted from the ASC-patches. Osteocalcin (osteoblasts), FABP4 (adipocytes) and aggrecan (chondrocytes) staining are shown.

**Figure 2 biomedicines-11-00987-f002:**
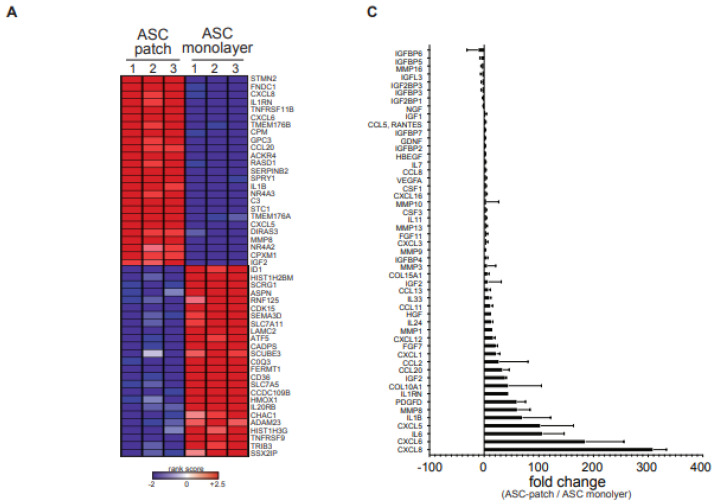
ASCs within the ASC-patch enhanced their regenerative transcriptome. The transcriptome of ASC in gelatin generated from 3 independent donors were assessed by microarray and compared to ASC from the same 3 donors grown in monolayer. (**A**) Heatmap of the top 25 genes with increased expression and the top 25 with decreased expression in the ASC-patch compared with ASC monolayer condition, ranked by Gene Set Enrichment Analysis (GSEA): The regulations considered to be significant were defined by the calculation of a fold change (> or <2) associated with a significant *p* value (<0.05). (**B**) Enrichment pathways determined using the Gene Ontology Biological Process (GOBP) gene set from the Molecular Signature Database are visualized. Heatmap of the top 25 genes with increased expression and the top 25 with decreased expression in SSc EEs compared to healthy donor (HD) EEs, ranked by Gene Set Enrichment Analysis (GSEA). Red dots: up-regulated, blue dots: down-regulated (**C**) Fold change analysis on 215 selected genes involved in wound healing. The 51 statistically significant regulations are shown (fold change > 2 (up) and <2 (down) and *p* < 0.05, non-parametric Mann–Whitney *t* test). Error bars refer to SD.

**Figure 3 biomedicines-11-00987-f003:**
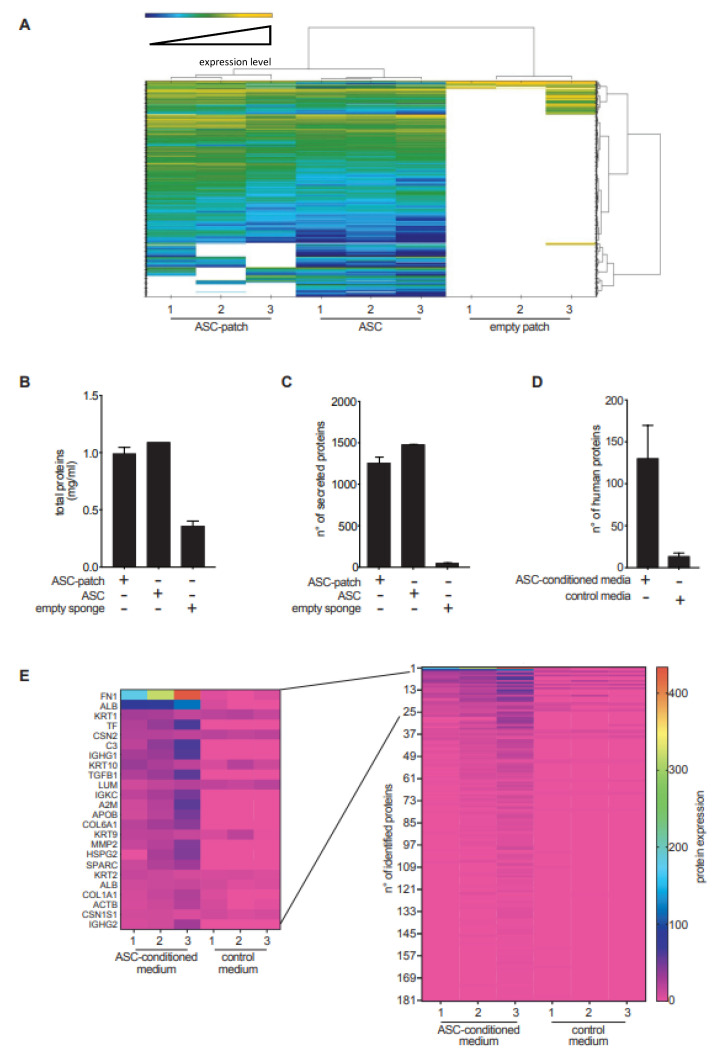
The ASC-patch concentrate of the ASC secretome and gelatin-derived products. (**A**–**C**) ASC-patches, ASC monolayer or empty gelatin sponge were incubated in a serum-free medium prior to a mass spectrometry analysis of the released proteome. ASC from three independent donors were analyzed. Hierarchical clustering analysis (**A**), total protein concentration (**B**) and total number of proteins identified (**C**) in each condition are shown. (**D**,**E**) An empty gelatin sponge was pre-incubated with serum-free media preconditioned or not with ASC. After enzymatic digestion of the gelatin sponge, a proteomic analysis was performed by mass spectrometry. (**C**) Total number of proteins identified by mass spectrometry in each condition. (**D**) Heatmap showing the relative quantification of the proteins as shown in C, with a zoom on the top-25 most abundant. Overall, in the figure, bars refer to mean ± SD.

**Figure 4 biomedicines-11-00987-f004:**
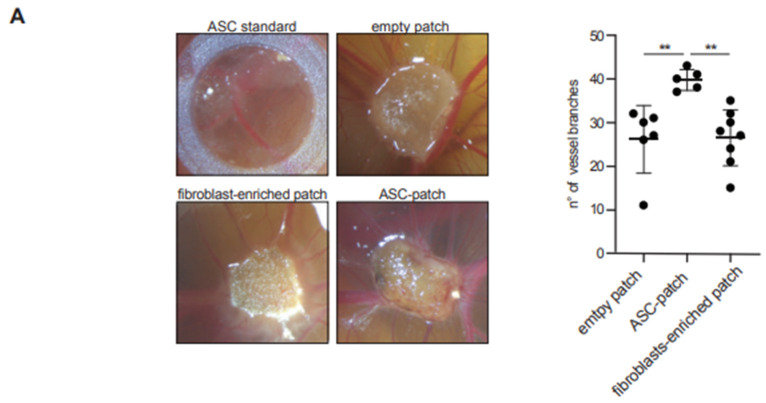
ASC-patch promotes neo-angiogenesis. (**A**) ASC-patches, fibroblast-containing patches, ASCs in single cell suspension (via a silicon ring) or empty gelatin sponges were deposited on the chorioallantoic membrane of fertilized chick eggs. Representative images of neovascularization are shown for each condition (left). The number of new vessels connections were quantified using ImageJ (right) (n = 6 assay). (**B**) A cell suspension of HUVEC was introduced in a Boyden chamber. Migration thorough a fibronectin-coated hemipermeable membrane towards the indicated conditioned media was determined (n = 5 assays). (**C**) Assessment of tubulogenesis of HUVEC in the presence of the indicated conditioned media. Tube area was quantified by the Cytation 5 image analysis software. (n = 6 assay) (**D**) Representative image of tube formation in the indicated conditioned media. Overall, in the figure: mean ± SD; non-parametric Mann–Whitney *t* test: * = *p* < 0.05, ** = *p* < 0.01.

**Figure 5 biomedicines-11-00987-f005:**
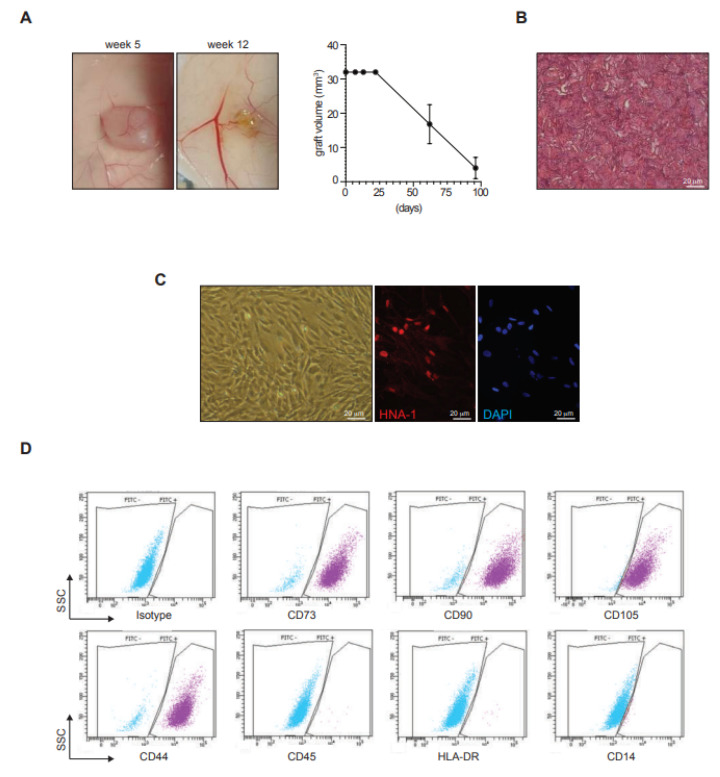
ASC-patches are stable in vivo after transplantation and do not show tumorigenicity. Ten Nu/Nu immunodeficient mice were transplanted subcutaneously with ASC-patches. Groups of 5 mice were sacrificed at weeks 5 and 12 for a macroscopic evaluation of transplants. (**A**) representative images of transplanted ASC-patches at week 5 and 12 (left) and quantification of graft volume over time (right). (**B**) Hemalun/eosin coloration of a transplanted ASC-patch at week 5. (**C**,**D**) Cells were extracted after dissociation of grafted ASC-patches at week 5. (**C**) Representative images of the morphology (left), and Human Nuclear Antigen (HNA)/DAPI immunofluorescence (middle and right) of the cells extracted from the transplanted ASC-patches. (**D**) representative flow cytometry analysis of the extracted cell line for the indicated ASC markers (positive: CD73, CD90, CD105; negative: CD45, HLA-DR, CD14).

**Figure 6 biomedicines-11-00987-f006:**
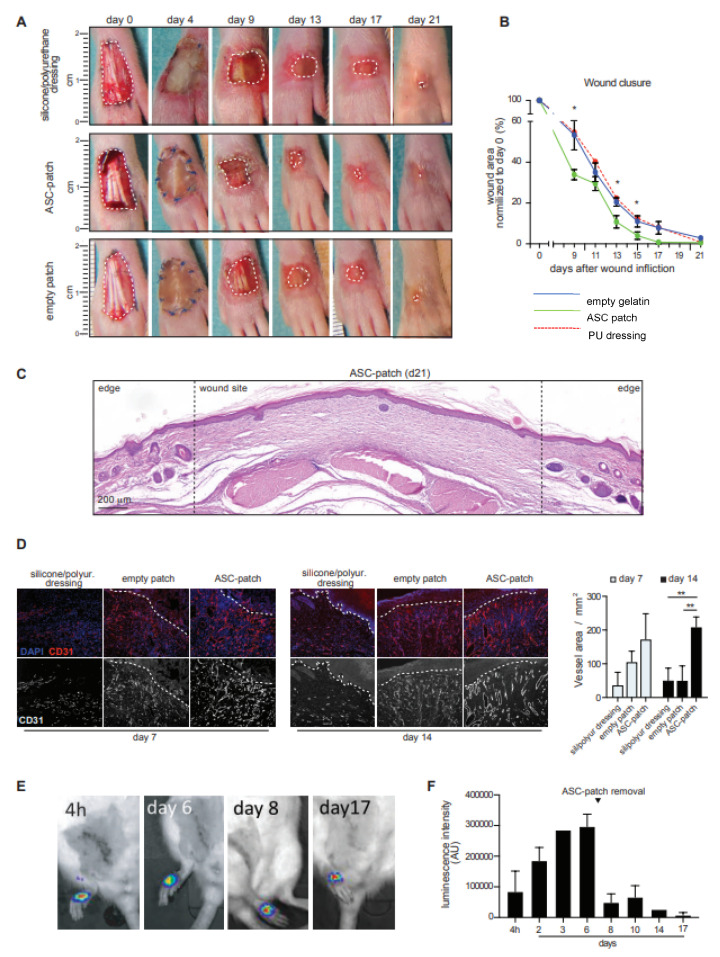
ASC-patches close ischemic wounds faster than acellular gelatin sponge and silicon/polyurethane dressings, with an increased vascularization of the granulation tissue. ASC-patches, empty gelatin sponge or a standard dressing of silicone/polyurethane were sutured on a full thickness wound on the ischemic foot of Wistar rats. Treatment was kept in place for the initial 7 days. (**A**) Representative macroscopic images are shown at different time points. (**B**) Quantification of the wound area, as normalized to day 0. Shown are cumulative data from 6 ASC-patches and 5 empty patches. A representative silicone/polyurethan dressing is shown as control. (**C**) Representative HE staining of a wound treated with the ASC-patch at day 21. (**D**) Endothelial cell marker CD31 (red—top panel; grey—bottom panel) and DAPI (blue, nuclei) immunofluorescent staining of the granulation tissue at day 7 and 14 (left). Quantification of the vessel area (defined by CD31 staining)/mm^2^ (right). (**E**) Representative live imaging of rASC _EFS FLuc_ after intraperitoneal injection of D-Luciferin. Images are taken one hour after the substrate injection. (**F**) Quantification of the luminescence emitted in the wound zone over 17 days post-treatment. Cumulative data from 3 rats is shown. Mean ± SD; non-parametric Mann–Whitney *t* test: ** = *p* < 0.01; * *p* < 0.05. AU: Arbitrary Units.

## Data Availability

Data are available on request from the authors.

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
