# Peer review of "Adipose-Derived Stromal Cells within a Gelatin Matrix Acquire Enhanced Regenerative and Angiogenic Properties: A Pre-Clinical Study for Application to Chronic Wounds"

_biomedicines, 2023, doi:10.3390/biomedicines11030987_

Round 1

Reviewer 1 Report

This manuscript addresses a study that evaluates the use of a cross-linked porcine gelatin sponge as an optimal carrier for Adipose-derived Stromal Cells (ASC) to regulate the regenerative activity of ASC for the delivery of ischemic wounds.. Globally this is a good, well-structured, and scientifically sound manuscript and could positively contribute wound healing field.

Below I point out some comments and issues that the author should consider for this manuscript:

1.        The abstract could be further improved by providing more technical details on the methodology and results. Additionally, the abstract could be revised to include more specific and quantitative details, such as the degree of upregulation of wound healing genes, the magnitude of the effect on endothelial cell migration and tubulogenesis, and the extent of angiogenesis observed in the chick chorioallantoic membrane model. Including these details could help the reader better understand the study's findings' significance and potential clinical impact.

2.        Abbreviations should be checked in the whole document to see if they have been spelled out first or explained;

3.        The introduction paragraph that discusses the limitations of current ASC delivery methods could be expanded to include more specific examples and details, such as the risks and complications associated with injection-based protocols. Please, give more specific references to previous studies, including both in vitro and in vivo work, to support the claims made in the text. The last sentence of the introduction could be rephrased to better reflect the study's goals and conclusions rather than just stating that the approach is "validated" and "paves the way" for further trials.

4.        The Materials and Methods section could benefit from some improvements, starting by explaining why the methods were chosen. Moreover, the section mentions that the ASC lines used in the study were fully validated for their phenotype, multipotency, and regenerative potential; it would be helpful to provide more information about how this validation was done and what criteria were used. It would also be helpful to provide more detail about statistical tests, significance levels, how many replicates were used for each experiment and how data were analyzed.

5.        Results and discussion sections show excellent outputs from the laboratory work. It is not clear whether any statistical tests were performed to determine whether the observed differences between ASC grown in monolayer and ASC grew in the gelatin sponge were statistically significant. If statistical tests were not performed, adding them would be a good idea. It might be useful to provide more information about the specific genes and pathways that were upregulated and downregulated in ASC grown in the gelatin sponge, and how these relate to the process of tissue regeneration. More recent references will be valuable.

The authors have done good laboratory work corresponding to the manuscript's global quality.

Author Response

  1. The abstract could be further improved by providing more technical details on the methodology and results. Additionally, the abstract could be revised to include more specific and quantitative details, such as the degree of upregulation of wound healing genes, the magnitude of the effect on endothelial cell migration and tubulogenesis, and the extent of angiogenesis observed in the chick chorioallantoic membrane model. Including these details could help the reader better understand the study's findings' significance and potential clinical impact.

Thank you for the helpful comment regarding the importance of the information in the abstract. All points suggested by the reviewer have been modified in the revised manuscript.

  1. Abbreviations should be checked in the whole document to see if they have been spelled out first or explained;

OK, it has been done

  1. The introduction paragraph that discusses the limitations of current ASC delivery methods could be expanded to include more specific examples and details, such as the risks and complications associated with injection-based protocols. Please, give more specific references to previous studies, including both in vitro and in vivo work, to support the claims made in the text. The last sentence of the introduction could be rephrased to better reflect the study's goals and conclusions rather than just stating that the approach is "validated" and "paves the way" for further trials.

Admittedly, a major challenge and thus a prerequisite for this study is the current lack of control/knowledge (and thus standardization) of the best mode of administration. Thus, as suggested by the reviewer, a more detailed description of the current limitations of administration, including injections, has been included in the introduction to the revised manuscript. In addition, and as requested, additional references have been added to support the work described in vitro and in vivo: references on preclinical knowledge of ASC for wound care (PMID 36835295, 33256038, 36571216), references highlighting the importance of routes of administration (PMID 3346840, 30118878). As requested, the last sentence of the introduction has been rephrased to better reflect the study's goals and conclusions.

  1. The Materials and Methods section could benefit from some improvements, starting by explaining why the methods were chosen. Moreover, the section mentions that the ASC lines used in the study were fully validated for their phenotype, multipotency, and regenerative potential; it would be helpful to provide more information about how this validation was done and what criteria were used. It would also be helpful to provide more detail about statistical tests, significance levels, how many replicates were used for each experiment and how data were analyzed.

Accordingly, a brief explanation of why the method was chosen has been added in the revised manuscript: microarrays, CAM model, HUVEC migration and tubulogenesis, tumorigenicity assay, rat model of ischemic wounds. Thank you for the comment asking to add how the ASC were validated. The new results section has added the two validation criteria (detailed phenotype and multipotency) according to the international standards. A reference to the international standard was also added. The request for statistics was also added where it was missing from the materials and methods and figure legends.

  1. Results and discussion sections show excellent outputs from the laboratory work. It is not clear whether any statistical tests were performed to determine whether the observed differences between ASC grown in monolayer and ASC grew in the gelatin sponge were statistically significant. If statistical tests were not performed, adding them would be a good idea.It might be useful to provide more information about the specific genes and pathways that were upregulated and downregulated in ASC grown in the gelatin sponge, and how these relate to the process of tissue regeneration. More recent references will be valuable.

This comment highlights that the issue of statistics needs to be presented more explicitly when 2D ASC are compared to gelatin sponge / ASC, especially in transcriptomics experiments. Transcriptomic was performed in biological triplicates: To ensure statistical relevance, 3 independent ASC lines from 3 independent donors were compared between the monolayer and gelatin sponge conditions. Figure 2C shows the most important and statistically significant regulations between 2D and gelatin sponge. Indeed, a fold change was calculated between the 2 conditions, as well as an FDR p-value, to keep only the regulations that are significant, i.e. with a fold change > or<2 and an FDR <0.05. Statistics are in conclusion included in this comparison, but not sufficiently explained in a clear manner. We have clarified the statistical analysis of the comparison in the revised manuscript (new legend for Figure 2C). 

The authors have done good laboratory work corresponding to the manuscript's global quality.

Authors thank the reviewer for this comment.

Reviewer 2 Report

1. The presented paper evaluates the healing influence of a gelatin sponge on Adipose-derived Mesenchymal Stromal Cells and disclose that this combination showed enhanced regenerative and angiogenic properties.

2. Introduction: In fact, cell monotherapy alone for wound healing may not show sufficient results and should use supporting biocompatible materials [ https://doi.org/10.1007/s12015-022-10379-z ], like collagen or gelatin. It is interesting that even in the case of wound healing, the structure, properties, and morphological parameters of the wound healed differ from those of surrounding tissues [ https://doi.org/10.1089/wound.2021.0039 ].

3. Results: The mechanical properties of the gelatin used (elastic modulus, elongation at break) should be given. The mechanical properties of implanted materials have a direct impact on the effectiveness of wound healing [ https://doi.org/10.1039/C9RA04026A ]

4. Discussion:

4.1. The features and advantages of the proposed approach need to be disclosed in more detail.

4.2. The short-term assessment of angiogenesis is a controversial issue and requires additional justification. For example, already in the early term of cell therapy (up to 7 or 14 days), intensive vascular growth is associated with epithelialization stimulation of epithelialization [https://doi.org/10.3390/dj9090101 ; https://doi.org/10.1016/j.injury.2022.09.041 ].

4.3. Furthermore, complications, side effects, and adverse events of MSC-based cell therapies should also be discussed.

5. The amount of additional material is impressive and substantiates the results of the article.

Author Response

  1. The presented paper evaluates the healing influence of a gelatin sponge on Adipose-derived Mesenchymal Stromal Cells and disclose that this combination showed enhanced regenerative and angiogenic properties.

  1. Introduction: In fact, cell monotherapy alone for wound healing may not show sufficient results and should use supporting biocompatible materials [ https://doi.org/10.1007/s12015-022-10379-z ], like collagen or gelatin. It is interesting that even in the case of wound healing, the structure, properties, and morphological parameters of the wound healed differ from those of surrounding tissues [ https://doi.org/10.1089/wound.2021.0039 ].

This comment regarding the route of administration and its current limitations agrees with reviewer 1. As we have already responded to reviewer 1, we agree that a major challenge and thus a prerequisite for this study is the current lack of control/knowledge (and thus standardization) of the best mode of administration and we agree that injection of a cell suspension is not sufficient. A more detailed description of the current limitations of delivery, including injections, has been included in the introduction to the revised manuscript. The additional references suggested by reviewer 2 has also been added, notably to support this point.

  1. Results: The mechanical properties of the gelatin used (elastic modulus, elongation at break) should be given. The mechanical properties of implanted materials have a direct impact on the effectiveness of wound healing [ https://doi.org/10.1039/C9RA04026A ]

The authors note this very useful suggestion regarding the mechanical properties of the scaffold used. It is indeed desirable to develop a patch whose structure, functions and mechanical properties are similar to those of healthy skin and compatible with healing. A number of scaffolds have been widely used in the field of tissue engineering and all emphasize the importance of hydrophilicity, biodegradability and biocompatibility. Collagen is an important component of the skin and provides strength. Gelatin is a hydrophilic, biocompatible, and inexpensive collagen-derived product and was considered to have the desired characteristics to promote ASC survival, adhesion, and activity. Based on the suggested reference (which has been added in the revised manuscript), these aspects of the biophysical and mechanical properties of gelatin have been added to the beginning of the results section of the revised manuscript.

  1. Discussion:

4.1. The features and advantages of the proposed approach need to be disclosed in more detail.

We agree with this suggestion and features /advantages of the proposed approach has been added in the revised discussion of the new manuscript

4.2. The short-term assessment of angiogenesis is a controversial issue and requires additional justification. For example, already in the early term of cell therapy (up to 7 or 14 days), intensive vascular growth is associated with epithelialization stimulation of epithelialization [https://doi.org/10.3390/dj9090101 ; https://doi.org/10.1016/j.injury.2022.09.041 ].

We thank reviewer 2 for these two relevant studies, which link MSC, vascularization, and re-epithelialization. We have discussed this link in the revised manuscript and added these two interesting references.

4.3. Furthermore, complications, side effects, and adverse events of MSC-based cell therapies should also be discussed.

We agree with this suggestion and complications, side effects, and adverse events of the proposed approach has been added in the revised discussion of the new manuscript

  1. The amount of additional material is impressive and substantiates the results of the article.

Authors acknowledge reviewer 2 for this comment.